# Development of an interactive, agent-based local stochastic model of COVID-19 transmission and evaluation of mitigation strategies illustrated for the state of Massachusetts, USA

Alexander Kirpich[1,2], Vladimir Koniukhovskii[3], Vladimir Shvartc[3], Pavel Skums[4], Thomas A. Weppelmann[5], Evgeny Imyanitov[6], Semyon Semyonov[7], Konstantin Barsukov[3], Yuriy Gankin[7]*

1 Department of Population Health Sciences, Georgia State University, Atlanta, Georgia, United States of America, 2 School of Public Health, Georgia State University, Atlanta, Georgia, United States of America, 3 EPAM Systems, Saint Petersburg, Russia, 4 Department of Computer Science, Georgia State University, Atlanta, Georgia, United States of America, 5 Department of Internal Medicine, University of South Florida, Tampa, Florida, United States of America, 6 N.N. Petrov Research Institute of Oncology, Saint Petersburg, Russia, 7 Quantori, Cambridge, Massachusetts, United States of America

☯ These authors contributed equally to this work.
* yuriy.gankin@quantori.com

**Data Availability Statement:** The manuscript contains all the references. They are also

## Abstract

Since its discovery in the Hubei province of China, the global spread of the novel coronavirus SARS-CoV-2 has resulted in millions of COVID-19 cases and hundreds of thousands of deaths. The spread throughout Asia, Europe, and the Americas has presented one of the greatest infectious disease threats in recent history and has tested the capacity of global health infrastructures. Since no effective vaccine is available, isolation techniques to prevent infection such as home quarantine and social distancing while in public have remained the cornerstone of public health interventions. While government and health officials were charged with implementing stay-at-home strategies, many of which had little guidance as to the consequences of how quickly to begin them. Moreover, as the local epidemic curves have been flattened, the same officials must wrestle with when to ease or cease such restrictions as to not impose economic turmoil. To evaluate the effects of quarantine strategies during the initial epidemic, an agent based modeling framework was created to take into account local spread based on geographic and population data with a corresponding interactive desktop and web-based application. Using the state of Massachusetts in the United States of America, we have illustrated the consequences of implementing quarantines at different time points after the initial seeding of the state with COVID-19 cases. Furthermore, we suggest that this application can be adapted to other states, small countries, or regions within a country to provide decision makers with critical information necessary to best protect human health.

duplicated below. The code is available here: https://github.com/quantori/COVID19-MA-Transmission The input data are available here: https://www.mass.gov/info-details/archive-of-covid-19-cases-in-massachusetts.

**Funding:** The authors Vladimir Koniukhovskii (VK), Vladimir Shvartc (VS) and Konstantin Barsukov (KB) are employed by the commercial company EPAM Systems in Saint Petersburg, Russia. The authors Yuriy Gankin (YG) and Semyon Semyonov (SS) are employed by the commercial company Quantori in Cambridge, Massachusetts, United States. EPAM Systems provided support in the form of salaries for authors VK and VS, but did not have any additional role in the study design, data collection and analysis, decision to publish, or preparation of the manuscript. The specific roles of these authors are articulated in the 'author contributions' section. The authors worked on the publication at their leisure time unrelated to their job duties. Quantori provided support in the form of salaries for authors YG and SS and relevant publication fees. YG is a Chief Scientific Officer (CFO) of Quantori was an inspirer for the project due to COVID-19 situation and SS worked on the project under the supervision of YG. The specific roles of these authors are articulated in the 'author contributions' section. Alexander Kirpich (AK), Pavel Skums (PS), Thomas A. Weppelmann (TAW), and Evgeny Imyanitov (EI) received no funding for their work on the project. The specific roles of these authors are articulated in the 'author contributions' section.

**Competing interests:** The authors declare that no competing interests exist. Alexander Kirpich (AK), Pavel Skums (PS), Thomas A. Weppelmann (TAW), and Evgeny Imyanitov (EI) received no funding for the work on the project. The authors Vladimir Koniukhovskii (VK), Vladimir Shvartc (VS) and Konstantin Barsukov (KB) are employed by the commercial company EPAM Systems in Saint Petersburg, Russia. The authors Yuriy Gankin (YG) and Semyon Semyonov (SS) are employed by the commercial company Quantori in Cambridge, Massachusetts, United States. The commercial affiliations of authors VK, VS, KB, YG, and SS do not alter their adherence to PLOS ONE policies on sharing data and materials. The data and the source code are made publicly available which is outlined in the manuscript. Quantori provided support in the form of salaries for authors YG and SS and relevant publication fees. YG is a Chief Scientific Officer (CFO) of Quantori was an inspirer for the project due to COVID-19 situation and desire to contribute. This will serve the overall good

## Introduction

The epidemic of a novel coronavirus was first detected in the city of Wuhan in the Chinese province Hubei on December of 2019 [1–4]. Despite the unprecedented efforts from Chinese authorities including the complete lockdown of the entire city of Wuhan on January 22, 2020 the virus has rapidly spread to all continents except Antarctica. The World Health Organization (WHO) officially declared the coronavirus a global pandemic on March 11, 2020 [5], only three months after the first case was detected. The novel coronavirus is now officially named SARS-CoV-2 and the disease caused by it has been called COVID-19 [6] to distinguish from SARS-CoV and the corresponding severe acute respiratory syndrome (SARS) pandemic from 2003 [7, 8]. Despite the much lower case-fatality rate, SARS-CoV-2 has caused morbidity and mortality orders of magnitude higher than severe acute respiratory syndrome (SARS) and Middle East respiratory syndrome (MERS) combined [9]. As of May 13, 2020, more than 4.1 million infections have been reported worldwide, with more than 287, 000 deaths due to complications of COVID-19 [10]. As of May, 2020 there is neither an effective virus-specific treatment, nor Food and Drug Administration (FDA) approved vaccine available for SARS-CoV-2 [11–18]. As such, social distancing and quarantine are the only available measures to reduce the transmission and prevent overwhelming the capacity of existing healthcare systems. Social distancing in this context means practice of physical distancing between individuals to prevent transmissions. Quarantine involves more targeted actions and restrictions for those individual who are or are suspected to be infected with the goal of keeping them away from the others. Starting at the epicenter of Hubei province [19] in January, 2020, governments around the world have implemented society lockdown measures of varying degrees [20–24]. Since such measures remain the only available tools to control the spread of the pandemic, it is critical to understand the transmission dynamics of SARS-CoV-2 in the population. This would allow for the prediction of COVID-19 cases and deaths over time under different mitigation strategies, which could be implemented to reduce morbidity and mortality; as well as the allocation of limited resources to medical providers.

To achieve this goal, multiple approaches can be implemented that are typically driven by the quality and precision of the available data. The most commonly reported data for epidemics are the incidence of new cases and deaths represented as a time series over the fixed intervals (e.g. days or weeks) aggregated across multiple regions and reporting sources [25–27]. This aggregated data can be used for incidence curve reconstruction, modeling, and prediction when more detailed information about each infected individual is not available [28–32]. Such models are called compartmental models [33] where the study population is divided into groups while individuals within each group are assumed to have the same characteristics of interest (e.g. susceptible, infected, vaccinated, or immune). The compartmental models are formulated via a defined system of differential equations that allows for both deterministic and stochastic formulations to quantify the uncertainty of the model fit. Those models provide insight into the underlying epidemic dynamics and allow for the prediction of future trends in incidence under different transmission scenarios, including interventions such as social distancing, quarantines, and vaccinations. Another approach are the agent-based models [34–39], which utilize a synthetic population, that attempts to realistically represent the social interactions in time and space between individuals with different characteristics and infection status [36]. Compared to compartment models, agent-based models rely on detailed data about the study population, making them more computationally intensive while allowing for more realistic simulations of human transmission pathways.

An example of an agent-based model that has been developed to study influenza but has been successfully applied to study other respiratory diseases, including the efficacy of

and is aligned with Quantori's values, policies and goals.

quarantine measures for containment of SARS-CoV-2 is the FluTE model [35, 40] which was specifically adopted with the name Corvid [38]. The FluTE model is based on the assumption that the synthetic study population is parsed in social subgroups and the interactions between them are modeled at different community levels (household, neighborhood, work), age groups, time of the day, and other characteristics. In this work, we propose an agent based model that can be applied by regional governments to local epidemiological settings, using incidence data from Massachusetts, USA. The proposed model incorporates the infected individuals that are reported in the beginning of an epidemic for a given area and their personal characteristics such as the date the infection was confirmation, geographic location, and demographic characteristics. Thus, by using a limited amount of information, such as the populations of each zip code and number of reported cases, a robust simulation is generated with time series data of predicted disease morbidity and mortality. Furthermore, by incorporating different quarantine strategies, the reduction in the number of new cases and deaths can be estimated for each locale depending on their unique characteristics. Since this model uses limited input data that are publicly available and is implemented in the form of an interactive web application, we believe that this tool could represent a widely adaptable format for state and local governments and health officials to make informed decisions as they consider easing or ceasing mandatory quarantines. Once effective treatments and vaccines become available, this framework could also be used to allocate treatment resources and plan vaccination campaigns tailored to fit different geographic regions. The reported demographics data can also be depersonalized in accordance to HIPAA regulations [41] to make the use of the model versatile and not to violate the privacy of individuals.

## Materials and methods

### The model structure

The model stochastic simulations are generating the infected individuals at different times and stages. Those individuals are indexed by $k$ and have individual characteristics presented below:

$$Q_k = \left( (x_k, y_k), tinf_k, det_k, stg_k, age_k, rad_k, p_{cont(k)}, cont_k, R_{0(k)}, sever_k, dur_k, st_k(t) \right). \quad (1)$$

The characteristics of each individual $Q_k$ are determined and updated within the simulation process and have the following details:

- $(x_k, y_k)$—the Cartesian coordinates (in pixels) of the individual $Q_k$ that do not change within the simulation process after they are introduced;

- $tinf_k$—the time of infection onset for the individual $Q_k$ which is measured in relation to the simulation baseline time denoted as 0;

- $det_k$—the detection time variable measured in days that corresponds to a period from an infection acquisition until the proper diagnosis and reporting of individual $Q_k$;

- $stg_k = (stg_{1(k)}, stg_{2(k)}, stg_{3(k)})$—the vector of durations of three disease infection stages measured in days that characterize the infectivity of the given individual during those stages. It is assumed that $stg_{1(k)} + stg_{2(k)} + stg_{3(k)} = det_k$;

- $age_k$—the age of the individual $Q_k$ at the time of the infection onset;

- $rad_k$—the distance (in meters) up to which the individual $Q_k$ is able to infect the nearby individuals;

- $p_{cont(k)}$—the probability that during each day the individual $Q_k$ has any contacts which lead to new infections;

- $cont_k = (\mu_{cont(k)}, \sigma^2_{cont(k)})$—the individual-specific parameters that define the distribution of the number of successful infection transmissions to other individuals within a given day. This number is generated randomly for each day $t$, provided that the individual has any transmissions on the given day (according to the contact probability $p_{cont(k)}$).

- $R_{0(k)}$—the individual's reproduction number. This variable stores the number of individuals that is infected by $Q_k$ during the infection period $det_k$. The average of those reproduction numbers across individuals and simulations are used to estimate the basic reproduction number $\mathcal{R}_0$ [42] which is a characteristic of the entire epidemic;

- $sever_k$—the disease severity variable for the individual $Q_k$ that takes three values, where 1 corresponds to lethal, 2 corresponds to severe, and 3 corresponds to mild; the disease severity does not change for a given individual after it is determined randomly from a trinomial distribution;

- $dur_k$—the disease duration from the infection onset to cure (or death) in days, which is generated randomly based on the $sever_k$ parameter;

- $st_k(t)$—the status of the individual $Q_k$ at a given day $t$. The status of the individuals within the simulation is expected to change over time and is expected to take the following values:

  - $st_k(t) = 0$—the individual is detected based on the external information i.e. from the reported data that are used as the model input;

  - $st_k(t) = 1$—the individual is infected but has not been identified as such yet;

  - $st_k(t) = 2$—the individual has been infected and detected as such, which has also implied the individual's isolation (quarantine).

  - $st_k(t) = 3$—the individual has recovered and is immune;

  - $st_k(t) = 4$—the individual has deceased.

The asymptomatic infections are incorporated inside the model via three severities that are generated for each infected individual. This incorporation of asymptomatic infections into those with a "mild" severity status is made to improve the model computational tractability. Since the model parameters are calibrated based on the detected cases and the output of the model presents the detected cases as well the prediction abilities for detected and reported cases within the model are preserved. The severities are discussed in details in the manuscript supplement S1 Appendix. In the beginning of the simulations the model utilizes multiple local epidemic epicenters $\boldsymbol{E} = \{E_1, E_2, \ldots, E_{\mathcal{I}}\}$. Those epicenters serve as the model initial conditions and represent the introductory geographic points for the index cases that are introduced into the susceptible population. The epicenters can either correspond to the actual address coordinates for those places where the initial outbreaks were detected or to the centers of the corresponding aggregated geographic units. The latter may be the case, if either the exact infection acquisition locations are not known, or the privacy concerns prevent the inclusion of such data into the model. In the latter case the centers of the aggregated geographic units are taken as epicenters $E_i$ for each $i = 1, 2, \ldots, \mathcal{I}$.

The local epicenters in the model are defined by a pair of geographic coordinates (*Lat*, *Long*) and by an epicenter-specific region radius $R_i$ which is defined in meters. Therefore, for

$i = 1, 2, \ldots, \mathcal{I}$ the epicenter region is defined by a triplet:

$$E_i = ((Lat_i, Long_i), R_i). \tag{2}$$

The epicenter regions are defined from the surveillance epidemiological data [43]. As the initial conditions in addition to the local epicenters the model incorporates the areas of high density $\boldsymbol{P} = \{P_1, P_2, \ldots, P_{\mathcal{J}}\}$ for $j = 1, 2, \ldots, \mathcal{J}$, where each $P_j$ represents a large city or a densely populated area and which is also defined by a triplet:

$$P_j = ((Lat_j, Long_j), R_j). \tag{3}$$

In the model the reporting times (days) for the initial index cases for each epicenter $i$ precede the modeled epidemic starting time which corresponds to the baseline time slot $t = 0$. Therefore, the reported time slot indexes across the epicenters $E_i$ are denoted as $s = 1, 2, \ldots, \mathcal{S}$ with the corresponding times $\tilde{t}_1, \tilde{t}_2, \ldots, \tilde{t}_{\mathcal{S}}$. The earliest reported cases and their dates are used for the model input with indexes $s = 1, 2, \ldots, \tilde{\mathcal{S}}$ such that $\tilde{\mathcal{S}} < \mathcal{S}$ and the corresponding times $\tilde{t}_1, \tilde{t}_2, \ldots, \tilde{t}_{\tilde{\mathcal{S}}}$. The corresponding number of confirmed and reported infections for each local epicenter $E_i$ up to and including the time $\tilde{t}_{\tilde{\mathcal{S}}}$ for $s = 1, 2, \ldots, \tilde{\mathcal{S}}$ is denoted as $n_i(\tilde{t}_{\tilde{\mathcal{S}}})$. The corresponding set of infected and reported (i.e. with the status $st_k(t) = 1$) individuals across all epicenter is denoted as:

$$\boldsymbol{D} = \{\tilde{Q}_1, \tilde{Q}_2, \ldots, \tilde{Q}_{\mathcal{K}_{\tilde{\mathcal{S}}}}\}, \tag{4}$$

where $k = 1, 2, \ldots, \mathcal{K}_{\tilde{\mathcal{S}}}$ is the global index for initial cases across all times $\tilde{t}_1, \tilde{t}_2, \ldots, \tilde{t}_{\mathcal{S}}$ and $\mathcal{K}_{\tilde{\mathcal{S}}}$ is the total number of the initial *index cases* that is simulated within the model based on the input data. The tilde notation for $\tilde{Q}_k$-s in $\boldsymbol{D}$ emphasizes the link to the model input data. The newly infected individuals are generated spatially in relation to the population area centers in the beginning and later in relation to the previously infected individuals. The individual's coordinates are generated randomly using the distribution mechanism described in details in the manuscript supplement S1 Appendix.

The time index that corresponds to individual day within the model is denoted as $t$ and is equal to 0 at the model baseline. The simulation baseline time $t = 0$ corresponds to the latest reporting time $\tilde{t}_{\tilde{\mathcal{S}}}$ of the earliest reported cases that are used for the model input. The actual infection times for those index cases precede the selected baseline simulation time $t = 0$ due to the infectivity periods generated for those index cases prior to their reporting. The actual simulation starting time that accounts for the infectivity periods is denoted as $t = T_{min}$ and is smaller than the baseline time $t = 0$. This simulation starting time $t = T_{min}$ is generated within the model, while the baseline time $t = 0$ is defined by the data and is defined by the largest index within the set of calibration indexes $s = 1, 2, \ldots, \tilde{\mathcal{S}}$. The largest simulation time $t = T_{max}$ is determined by the model user based on the desired length of prediction. The initial set of index cases $\boldsymbol{D}$ from (4) defines the model initial cases that are allocated across the local epicenters (2) at times up to the baseline time $t = 0$.

Based on the model geographic characteristics (2) and (3) and the initial set of reported individuals $\boldsymbol{D}$ from (4) the new lists $\mathcal{L}(t)$ of of modeled individuals are simulated for time slots $t \in [T_{min}; T_{max}]$ where $T_{max} - T_{min} + 1$ is the total number of the simulated time slots. The simulated lists $\mathcal{L}(t)$ have the following format:

$$\mathcal{L}(t) = \{Q_1, Q_2, \ldots, Q_{\mathcal{K}(t)}\} \tag{5}$$

where the value of $\mathcal{K}(t)$ is defined by the simulation at every simulation time step $t \in [T_{min};$

$T_{max}$]. During this procedure the input set individuals $\boldsymbol{D}$ defined in (4) is allocated between the different epicenters and time slots within the lists $\mathcal{L}(t)$ defined by (5). The allocations of the set $\boldsymbol{D}$ is performed during the time slots $t \in [T_{min}; \tilde{t}_{\tilde{\mathcal{S}}}]$ where $\tilde{t}_{\tilde{\mathcal{S}}} < T_{max}$.

## The overall model flow

The entire modeling process can be summarized via the following steps:

- The model input time interval is determined by fixing the first $\tilde{\mathcal{S}}$ reporting indexes out of the total $\mathcal{S}$ where $\tilde{\mathcal{S}} < \mathcal{S}$. Those indexes correspond to the reporting time slots $\tilde{t}_1, \tilde{t}_2, \ldots, \tilde{t}_{\tilde{\mathcal{S}}}$. This completely defines the list of reported index cases $\boldsymbol{D}$ from (4) that are used as the model initial conditions. The baseline time of the model $t = 0$ is assumed to correspond to $\tilde{t}\tilde{\mathcal{S}}$.

- The individuals from the reported set $\boldsymbol{D}$ that are defined in (4) are assigned to the local epicenters of the future epidemic $E_i$ for $i = 1, 2, \ldots, \mathcal{I}$ based on the available (from the input data) geographic distribution.

- The geographic data about the areas of high density $P_j$ for $j = 1, 2, \ldots, \mathcal{J}$ are incorporated into the model.

- The model is initialized with the index cases from $\boldsymbol{D}$. Based on those index cases that are defined in (4) the initial infection time $T_{min}$ is determined. This step is necessary to incorporate the infection times that have been present before the first reporting time $\tilde{t}_1$ into the model.

- The final time point of the stochastic simulations $T_{max}$ is define by the user based on the desired study and prediction goals.

- The initial list of infected individuals $\mathcal{L}(T_{min})$ is initialized at time $T_{min}$ only with the earliest model input cases from the list $\boldsymbol{D}(T_{min})$.

- The infected list of individuals $\mathcal{L}(t + 1)$ for the time slot $t + 1$ is generated sequentially for all $t \in [T_{min}; T_{max} - 1]$ based on the list of individuals $\mathcal{L}(t)$ from previous time slot $t$ and the individual's characteristics within the list $\mathcal{L}(t + 1)$ are updated at this time step $t + 1$. The details of the new infection generations are provided in S1 Appendix.

Based on the lists $\mathcal{L}(t)$ at every time slot $t \in [T_{min}; T_{max}]$ the infected modeled population summaries can be computed and summarized. In particular, the total number of currently infected but not identified individuals (i.e. those with the status $st(t) = 1$) is saved into $Inf(t)$ variable for every $t$. The total number of treated or quarantined individuals (i.e. with the status $st(t) = 2$) is saved into $Treat(t)$ variable for every $t$. The total number of recovered individuals (i.e. with the status $st(t) = 2$) is saved into $Recov(t)$ variable for every $t$. The total number of deceased individuals (i.e. with the status $st(t) = 4$) is saved into $Dead(t)$ variable for every $t$. Those numbers are used in the model calibration procedures, epidemiological summaries and in the model predictions. The model input utilizes only the first $\tilde{\mathcal{S}}$ reported indexes with the corresponding reported times $\tilde{t}_s$ for $s = 1, 2, \ldots, \tilde{\mathcal{S}}$ with the total number of reported indexes equal to $\mathcal{S}$ and $\tilde{\mathcal{S}} < \mathcal{S}$. The remaining reported indexes $\tilde{\mathcal{S}} + \tau, \tilde{\mathcal{S}} + \tau + 1, \ldots, \mathcal{S}$ for some integer $\tau$ are divided into the two groups:

$$\{\tilde{\mathcal{S}} + \tau, \tilde{\mathcal{S}} + \tau + 1, \ldots, \dot{\mathcal{S}}\} \text{ and } \{\dot{\mathcal{S}} + 1, \dot{\mathcal{S}} + 2, \ldots, \mathcal{S}\}. \tag{6}$$

The first group of the reported indexes from (6) is used for the model calibration and estimation of the unknown parameters. The second group of the reported indexes from (6) is

used to evaluate the quality of the model predictions. The Massachusetts surveillance data that are used for the model calibration, validation and predictions are freely available at the Massachusetts Department of Public Health web site [43]. The first reported date which corresponds to the time index $\tilde{t}_1$ in the model is March 13, 2020. The latest reported date that is used for the model input is March 26, 2020 which corresponds to the time index $\tilde{t}_{\hat{S}}$ in the model. The time indexes that correspond to $\tilde{t}_{\hat{S}+\tau}$ and $\tilde{t}_{\hat{S}}$ are April 14, 2020 and April 22, 2020 respectively. The parameter optimization is performed by minimizing the sum of squared differences between the model-produced outputs and the calibration data by using the Nelder–Mead numerical minimization method [44]. The additional details about the model formulation, parameterization, and calibration are summarized in the S1 Appendix.

After the model calibration is performed various quarantine and transmission intervention strategies are investigated within the calibrated model. Those intervention strategies are based on the assumption that the probability of contacts between the individuals decreases after the quarantine measures are enforced. Within the model this is implemented by an immediate change in the contact probability parameter starting from a certain calendar date. The earlier implementation of the quarantine in the model is represented by an earlier calendar date when the change in parameter value occurs. Those earlier quarantine dates result in the smaller transmission probability and fewer infected individuals in comparison to the later quarantine dates. The quarantine date in the end of the simulation period corresponds to no quarantine scenario. As an example, three scenarios with different quarantine start dates are presented and the corresponding reduction in the number of cases is discussed. There is also an option to consider other quarantine enforcement dates interactively within the current model implementation.

## The model availability

The model has been implemented in multiple environments which include the application tool for Microsoft Windows [45] and the web prediction tool [46, 47] (summaries only). The model application tool for Microsoft Windows is freely available under the terms of the MIT license [48]. The tool source code, the application, and the relevant documentation are available on GitHub [45]. The current model tool has been calibrated based on the state of Massachusetts (United States) incidence data [43]. In addition to that the user has an option to adjust interactively the tool parameters which include, in particular, the transmission parameters and the quarantine implementation dates(s). Overall, the proposed framework and the code are fairly general and can be adopted for other areas and territories where the demographics of the incidence cases and population characteristics are known with at least some geographic precision, and where the rapid evaluations of social distancing measures have to be quantified.

## Results

Within the model multiple epidemic progression scenarios can be considered. In particular, three different quarantine strategies are presented in this manuscript as an illustration of the model. The alternative quarantine scenarios can be produced and customized interactively within the model application tool if necessary. The difference between the presented scenarios is in the quarantine date at which the transmission probability parameter changes to smaller one. The smaller transmission parameter values result in fewer infections therefore the earlier quarantine dates result in less infections overall in comparison to the later dates. The predicted numbers of infections for each scenario are defined by the quarantine implementation date and those number can be compared. The first scenario corresponds to the quarantine date on

**Table 1. The predicted number of cumulative cases produced by the model over time for three different quarantine scenarios and three time periods together with the corresponding 90% prediction intervals.**

| Scenario | Quarantine Date | April 26, 2020 | May 26, 2020 | June 26, 2020 |
|---|---|---|---|---|
| First | March 29, 2020 | 24, 039 (20, 665;27, 296) | 32, 692 (27, 361;38, 221) | 36, 767 (30, 288;43, 976) |
| Second | April 06, 2020 | 56, 587 (46, 944;66, 401) | 89, 727 (72, 843;106, 797) | 105, 464 (84, 859;127, 796) |
| Third | April 13, 2020 | 123, 351 (100, 113;144, 018) | 245, 255 (197, 748;294, 750) | 307, 128 (243, 184;362, 104) |

March 29, 2020 i.e. the early reduction in contact probabilities and social distancing between individuals. The second scenario assumes the implementation of the quarantine measures on April 6, 2020, and the third scenario assumes the implementation of the quarantine measures on April 13, 2020. The point estimates and the prediction bands have been produced by replicating each of the three scenarios and taking the median values across 500 model runs for the point estimates and 5-th and 95-th percentiles for the 90% prediction intervals. The results are summarized in Tables 1 and 2 for the model-predicted cases and deaths, respectively. For example, the summaries from Table 1 can be compared after one month of the baseline date i.e. on April 26, 2020. For the first scenario the model predicts 24, 039 cumulative cases (with the 90% PI (20, 665;27, 296)), for the second scenario the model predicts 56, 587 cumulative cases (with the 90% PI (46, 944;66, 401)), and for the thirds scenario the model predicts 123, 351 cumulative cases (with the 90% PI (100, 113;144, 018)). Compared to the quarantine start date in the second scenario, the first scenario results in 58% reduction in cumulative cases on April 26, 2020, in 63% reduction in cumulative cases on May 26, 2020, and in 65% reduction in cumulative cases on June 26, 2020. Compared to the quarantine start date in the third scenario, the first scenario results in 81% reduction in cumulative cases on April 26, 2020, in 87% reduction in cumulative cases on May 26, 2020, and in 88% reduction in cumulative cases on June 26, 2020. Based on the model outputs the earliest quarantine measures and the reduction in contact probabilities can be extremely beneficial in mitigation of the outbreak consequences. The analogues summaries for the model-predicted death across the three scenarios are summarized in Table 2. Compared to the quarantine start date in the second scenario, the first scenario results in 52% reduction in cumulative deaths on April 26, 2020, in 63% reduction in cumulative deaths on May 26, 2020, and in 65% reduction in cumulative deaths on June 26, 2020. Compared to the quarantine start date in the third scenario, the first scenario results in 70% reduction in cumulative deaths on April 26, 2020, in 86% reduction in cumulative deaths on May 26, 2020, and in 88% reduction in cumulative deaths on June 26, 2020.

The model is presented via the graphic user interface (GUI) application for Microsoft Windows as well as the as web prediction tool [46, 47] (summaries only) that can be used for the geographic visualization of various epidemiological curves and geographic visualization of cases in the state of Massachusetts. Users can work with the tool and utilize the available model customizations. The appearance of the GUI for MS Windows application is presented in Fig 1. The state of Massachusetts MassGIS data were used to produce the tool map. The data

**Table 2. The predicted number of cumulative death produced by the model over time for three different quarantine scenarios and three time periods together with the corresponding 90% prediction intervals.**

| Scenario | Quarantine Date | April 26, 2020 | May 26, 2020 | June 26, 2020 |
|---|---|---|---|---|
| First | March 29, 2020 | 1, 432 (1, 248;1, 614) | 2, 236 (1, 879;2, 619) | 2, 603 (2, 156;3, 066) |
| Second | April 06, 2020 | 2, 959 (2, 473;3, 405) | 5, 987 (4, 863;7, 118) | 7, 397 (5, 947;8, 870) |
| Third | April 13, 2020 | 4, 813 (3, 996;5, 591) | 16, 046 (12, 859;19, 007) | 21, 339 (16, 884;25, 086) |

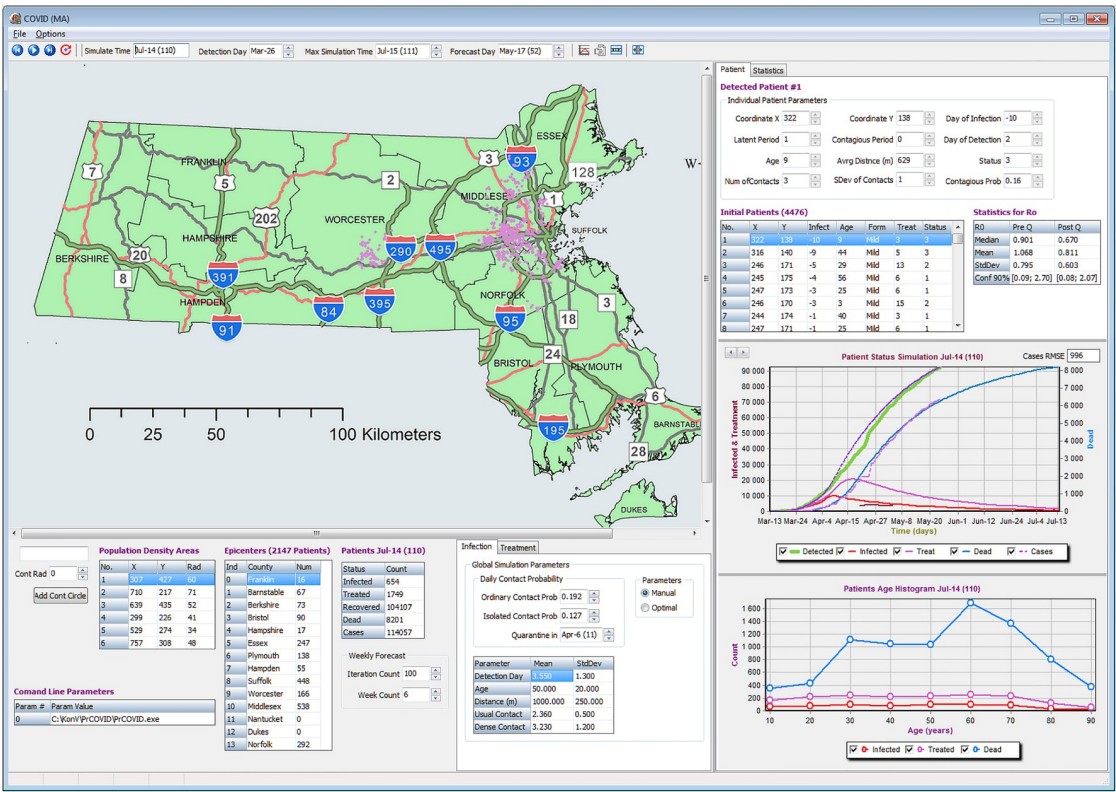

**Fig 1. The tool graphic user interface.** The visualization tool GUI for Microsoft Windows. The tool provides the geographic visualizations of epidemic on the state of Massachusetts map and constructs the epidemiological summaries and curves. The tool allows interactive model calibrations and step-by-step simulations.

are available for download and are public records from the Bureau of Geographic Information (MassGIS), Commonwealth of Massachusetts, Executive Office of Technology and Security Services [49].

The presented application does not require installation and can be launched directly by running the executable file. The tool allows the parameters optimizations and the visual animated simulations of the model outputs and their comparison with the reported data. The user can interactively customize the most important simulation parameters, change the duration of prediction, and adjust manually the locations of high population density. This allows the user to consider multiple scenarios of the epidemic spread. In addition, the user has flexibility to re-run the model multiple times either step by step or entirely for all time slots. The user can scroll via each day of the recently completed simulation to see the visualized results of that specific day and choose which epidemiological curves to include in the summary graphs. The modeled cases and other summaries are saved into the comma-separated values (csv) files after the end of each simulation. The tool also provides the estimate of the population basic reproduction number $\mathcal{R}_0$ for each simulation run together with the corresponding 90% confidence intervals. The estimates for $\mathcal{R}_0$ are provided based on the quantiles of the individual's reproduction numbers $R_{0(k)}$ across multiple individuals $k$ both before and after the quarantine date that is defined by the user. The resulting distribution of the individual's reproduction numbers $R_{0(k)}$ from the tool before and after the quarantine are summarized in Fig 2.

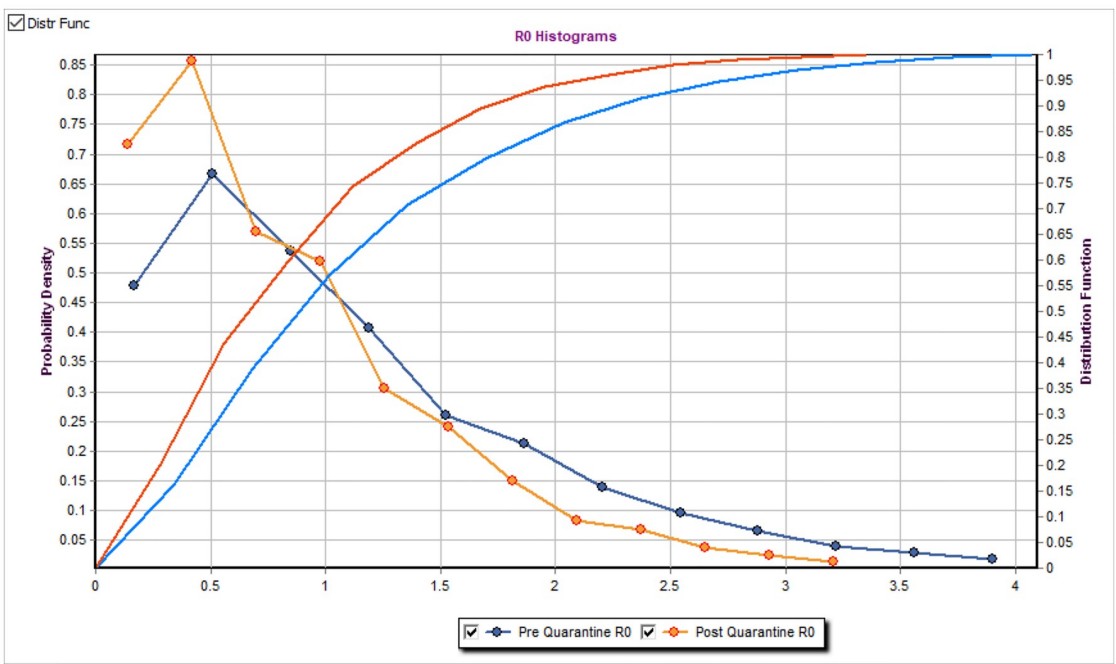

**Fig 2. The distribution of the individual's reproduction numbers $R_{0(k)}$.** The example of the tool output for the distribution of the individual's reproduction numbers $R_{0(k)}$. The output graph contains the estimated probability density function of the individual's reproduction numbers $R_{0(k)}$ together with the cumulative distribution functions both before and after the quarantine implementation date.

The example of the summary graphs for the model-produced outputs for the second scenario from Tables 1 and 2 are presented in Fig 3, which contains the four combined graphs available in the "Statistics" tab in the top right corner of the tool. Those graphs within the tool can be produced by setting the "Max Simulation Time" and "Forecast Day" fields to July 15, 2020 and by running the model 500 times by using the "Daily Forecast Evaluation" button. The 500 runs are necessary to produce the median predictions and the corresponding 90% uncertainty prediction bands across those runs by taking the 5-th and the 95-th percentiles across those situations for each time slot. Those graphs include the cumulative numbers of reported cases and deaths, together with the currently hospitalized patients and unreported cases. The graphs also include the reported data in blue. The calibration time period is highlighted in blue and is bounded by vertical bars.

## Discussion

In this work the local agent-based modeling framework for respiratory diseases has been presented. This framework incorporates the reported geographic incidence data that are typically available from surveillance, which include individual's age, infection status, and the severity of the disease. The model accounts for the latent period of the individual's infection before detection and proper reporting as well as for different disease severity levels. The framework also allows to incorporate the exact geographic addresses of individuals (if available) or the random geographic distribution of individuals within those aggregated districts where they are reported in case of privacy concerns. The model allows to perform predictions with different levels of social isolation between individuals and quarantine measures. Those measures are implemented at different times to compare different quarantine scenarios. As expected, there

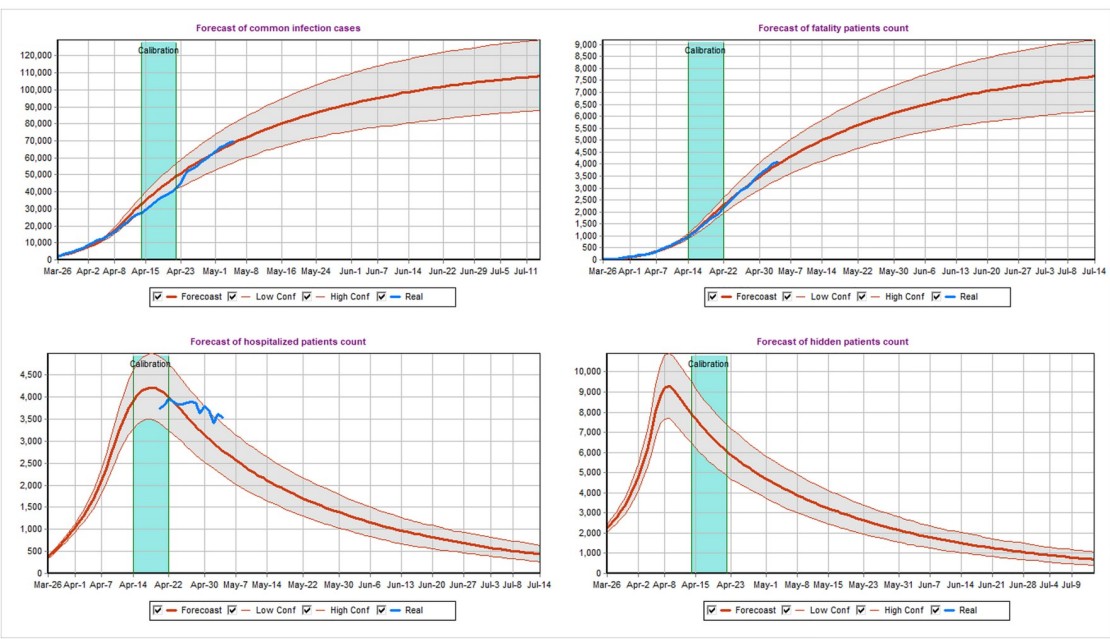

**Fig 3. The model-produced predictions.** The median of the model-produced 500 runs together with the corresponding 90% uncertainty prediction bands for different model outputs. The top left graph includes the cumulative numbers of reported cases. The top-right graph summarizes the cumulative subset of the reported cases that have deceased. The bottom graphs summarize the number of hospitalized and unreported patients in the given moment of time. The reported data are displayed in blue for visual comparison.

was a decrease in the cumulative incidence and deaths inversely proportional to the date quarantine was implemented; which resulted in approximately 50-80% reduction in cases and deaths depending on the scenario. This agrees with already published results that strict social distancing combined with proper testing will keep the disease at level that does not overwhelm the capacity of heath care system [18].

Compared to complex agent-based models, the compartmental models are based on the assumptions of homogeneous mixing and can be parameterized by a relatively small set of rates and initial conditions. The main challenges for the compartmental models [30, 31, 50] are the determination of the compartment types that are used in the model, the assignment of individuals between compartment i.e. the specifications of the set of rules that assign each particular individual to each type of compartments, and the determination of the parameters of interest which can either be postulated from external sources or estimated from data.

The agent based models, due to their inherited complexity, incorporate separate individuals with multiple different characteristics and parameters per individual. This adds another layer of parameterization flexibility, but also introduces another layer of modeling challenges, since the number of individual's characteristics within the model is determined by the modeler [18, 34–36]. This typically implies some additional assumptions about the model behavior and the overall model parametrization. Those assumptions are expected to be region-specific, since the transmission times, patterns, and other characteristics typically wary from region to region. The respiratory infections, which include SARS-CoV-2, add another level of complexity to any model due to quality of reported data. The asymptomatic cases are typically neither identified nor properly recorded and surveillance systems for symptomatic cases are never perfect either. As a result a substantial (but unknown) amount of cases is not reported and the modeler has

to account for those asymptomatic and unreported symptomatic cases who still participate in the disease transmission process.

Ideally, the model has to be: 1) flexible enough to incorporate the possible social and geographic characteristics of individuals and to provide the way to realistically represent the social interactions and the disease transmission mechanisms; 2) simple enough to avoid the problems with parameterization, but able to capture the actual transmission patterns with the goal of predictions and intervention studies; 3) utilize the available surveillance and public health data in the best possible way. The best possible way in this context means, that all the information from the data that can be used to answer the questions of interest are utilized, while the number of assumptions within the model beyond the information available from the data is the smallest possible that is necessary to implement the model.

In the case of COVID-19, an epidemic which has quickly evolved into a pandemic, the local epidemic developments in every region are expected to have different dynamics influenced by multiple region-specific factors. Thus, an agent based model which utilizes local settings is likely superior to a global agent-based model in this setting and can be implemented with minimal inputs as long as local data are available. In this example, we chose regional data for the state of Massachusetts, however we believe this framework and interactive tool could be adopted and useful for small or middle size countries or other administrative districts within a larger country, that have comparable reporting and data quality across different administrative regions.

## Conclusion

In this paper, we have presented a novel, localized agent-based model that can be used within minimal input data, which is publicly available and tailored to the population distributions of Massachusetts, USA. After calibration the model provided a good estimation of the actual incidence, hospitalizations, and death rates, with the added benefit of estimating the number of undetected infections in the population. Given the necessity for making decisions of easing or ceasing quarantines that are specific to a state or county based on their reported case counts, adaptation of this framework could prove to be very useful with efforts to reopen the economy, while quantifying the disease burden posed by such decisions. In addition, this model could be used for future outbreaks of other novel respiratory diseases to protect public health and possibly designed tailored interventions of treatment and vaccination campaigns.

## Supporting information

**S1 Appendix. Model details.** The details about the model formulation, parameterization, and calibration.
(PDF)

## Author Contributions

**Conceptualization:** Alexander Kirpich, Vladimir Koniukhovskii, Vladimir Shvartc, Pavel Skums, Thomas A. Weppelmann, Evgeny Imyanitov, Yuriy Gankin.

**Data curation:** Vladimir Shvartc, Semyon Semyonov, Konstantin Barsukov.

**Formal analysis:** Vladimir Koniukhovskii.

**Methodology:** Alexander Kirpich, Vladimir Koniukhovskii, Vladimir Shvartc, Pavel Skums, Evgeny Imyanitov, Yuriy Gankin.

**Project administration:** Semyon Semyonov, Yuriy Gankin.

**Resources:** Semyon Semyonov, Yuriy Gankin.

**Software:** Vladimir Koniukhovskii, Semyon Semyonov.

**Supervision:** Yuriy Gankin.

**Validation:** Alexander Kirpich, Vladimir Koniukhovskii.

**Visualization:** Alexander Kirpich, Vladimir Koniukhovskii, Pavel Skums, Thomas A. Weppelmann, Semyon Semyonov.

**Writing – original draft:** Alexander Kirpich, Vladimir Koniukhovskii, Vladimir Shvartc, Pavel Skums, Thomas A. Weppelmann, Evgeny Imyanitov, Yuriy Gankin.

**Writing – review & editing:** Alexander Kirpich, Thomas A. Weppelmann, Yuriy Gankin.

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
