## [Decision Letter · Decision Letter 0]

12 Oct 2020

PONE-D-20-24277

Development of an interactive, agent-based local stochastic model of COVID-19 transmission and evaluation of mitigation strategies illustrated for the state of Massachusetts, USA

PLOS ONE

Dear Dr. Gankin,

Thank you for submitting your manuscript to PLOS ONE. After careful consideration, we feel that it has merit but does not fully meet PLOS ONE’s publication criteria as it currently stands. Therefore, we invite you to submit a revised version of the manuscript that addresses the points raised during the review process.

We look forward to receiving your revised manuscript.

Kind regards,

Gergely Röst

Academic Editor

PLOS ONE

Journal Requirements:

We note that one or more of the authors are employed by a commercial company: EPAM Systems, Quantori.

2.1. Please provide an amended Funding Statement declaring this commercial affiliation, as well as a statement regarding the Role of Funders in your study. If the funding organization did not play a role in the study design, data collection and analysis, decision to publish, or preparation of the manuscript and only provided financial support in the form of authors' salaries and/or research materials, please review your statements relating to the author contributions, and ensure you have specifically and accurately indicated the role(s) that these authors had in your study. You can update author roles in the Author Contributions section of the online submission form.

2.2. Please also provide an updated Competing Interests Statement declaring this commercial affiliation along with any other relevant declarations relating to employment, consultancy, patents, products in development, or marketed products, etc.  

3. We note that Figure 1 in your submission contain map images which may be copyrighted. All PLOS content is published under the Creative Commons Attribution License (CC BY 4.0), which means that the manuscript, images, and Supporting Information files will be freely available online, and any third party is permitted to access, download, copy, distribute, and use these materials in any way, even commercially, with proper attribution. For these reasons, we cannot publish previously copyrighted maps or satellite images created using proprietary data, such as Google software (Google Maps, Street View, and Earth). For more information, see our copyright guidelines: http://journals.plos.org/plosone/s/licenses-and-copyright.

3.1.    You may seek permission from the original copyright holder of Figure 1 to publish the content specifically under the CC BY 4.0 license. 

3.2.    If you are unable to obtain permission from the original copyright holder to publish these figures under the CC BY 4.0 license or if the copyright holder’s requirements are incompatible with the CC BY 4.0 license, please either i) remove the figure or ii) supply a replacement figure that complies with the CC BY 4.0 license. Please check copyright information on all replacement figures and update the figure caption with source information. If applicable, please specify in the figure caption text when a figure is similar but not identical to the original image and is therefore for illustrative purposes only.

Reviewers' comments:

Reviewer's Responses to Questions

**Comments to the Author**

1. Is the manuscript technically sound, and do the data support the conclusions?

Reviewer #1: Partly

Reviewer #2: Yes

2. Has the statistical analysis been performed appropriately and rigorously? 

Reviewer #1: Yes

Reviewer #2: Yes

3. Have the authors made all data underlying the findings in their manuscript fully available?

Reviewer #1: Yes

Reviewer #2: Yes

4. Is the manuscript presented in an intelligible fashion and written in standard English?

Reviewer #1: Yes

Reviewer #2: Yes

5. Review Comments to the Author

Reviewer #1: Kirpich et al present an agent-based model of COVID spreading. They fit their model to data on COVID infections in Massachusetts. And provide an online tool to test their model.

There are a few major issues with the presentation and the results of the manuscripts:

1. The authors do not consider asymptomatic COVID patients in the model, but they could also spread the virus. So the fits provided only to symptomatic patients could be mis-parameterized due to the lack of the presence of asymptomatic patients in the model.

2. It is also a bit strange how virus spreading is handled. There are initial epicenters of the infections, but agents are not directly moving, rather just changing their contacts. The pandemic here spreads between places, but unclear how the spatial structure is taken into account – out of plotting the results on a map.

3. Most of the details of the model are presented only in the supplement and even there some details are unclear. For instance, the authors discuss the effects of quarantines and social distancing, but it is not clear how the model is adjusted to take into account these effects. They mention 3 quarantine scenarios, but it seems these are just three separate times of quarantine initiation. Based on the text, only individuals with symptoms are quarantined, which is a far weaker restriction than what is generally applied by authorities.

4. In the responses to earlier referees it is stated that various answers and extensions were added to the text, but unclear how the text was eventually changed. No supporting file of tracked changes is provided. There is one tracked file for the supplement, but it seems only half a sentence was changed in that document.

Reviewer #2: I have finished my review of the assigned manuscript. In this study, the authors develop a complex agent-based model to elucidate transmission dynamics of COVID-19. Their model includes unique features and substantial heterogeneity, not found in other compartmental based models.

1) In the agent based model, the location of the agents are given by pixel coordinates. However, it is unclear how many agents can occupy a particular pixel. Is it one agent/pixel? What about the resolution of the epicenters and population densities? Are they at the pixel resolution also?

2) The parameters used in the study should be summarized in the table. It is unclear how the natural history of disease progresses for each agent. For example, in the appendix, the authors use a Lognormal distribution to sample the durations of stg_{1, 2} in equations 8 and 9, but no reference or justification is given for why these durations were picked. (The authors' reply to another reviewer suggest they indeed included a table, but it should be provided in the main manuscript.)

4) Instead of justifying the parameter values used in their study, the authors (correctly) state that they are input values and interested readers have the ability to modify these values in the provided software. There are a few concerns with this:

--- the code is compiled only for Windows, leaving Mac and Linux users unable to run the software (without jumping through hoops such as virtual machines). For these users, the authors should justify their parameters and provide some sensitivity analysis.

--- If the goal is to only provide a framework/software, it should be provided as a web-based tool. Moreover, the narrative of the manuscript needs to be revised to indicate that the work simply provides a tool. As currently written, the narrative is as if they are trying to describe transmission dynamics of COVID-19. For this, the authors definitely need to provide more justification for parameter values.

--- The authors should check in a README.md file in their github repository with explicit instructions on compiling and running the software. In addition, all the comments (which describe the flow) are not in English, which makes it difficult to verify/find any possible bugs in the system.

6. PLOS authors have the option to publish the peer review history of their article (what does this mean?). If published, this will include your full peer review and any attached files.

Reviewer #1: No

Reviewer #2: No

---

## [Author Response · Author response to Decision Letter 0]

3 Nov 2020

Please find our detailed responses to reviewers as a part of our re-submission.

---

## [Decision Letter · Decision Letter 1]

11 Jan 2021

PONE-D-20-24277R1

Development of an interactive, agent-based local stochastic model of COVID-19 transmission and evaluation of mitigation strategies illustrated for the state of Massachusetts, USA

PLOS ONE

Dear Dr. Gankin,

Thank you for submitting your manuscript to PLOS ONE. After careful consideration, we feel that it has merit but does not fully meet PLOS ONE’s publication criteria as it currently stands. Therefore, we invite you to submit a revised version of the manuscript that addresses the points raised during the review process.

ACADEMIC EDITOR:

Dear Authors,

for the final version please take into consideration the following remarks of the reviewers.

"The authors properly answered my questions in their response letter. My major concern about asymptomatic patients is now mentioned in the main text, with a reference to more details in the supplement, but the supplement was only minimally changed and the word asymptomatic does not even appear in the supplement. Thus some further update on the supplement and maybe on the main text could help to make this complete.

The results and the discussion sections are relatively short. It might have been useful to add a section on the limitations of the model as it stands now.

The above points are quite properly explained now in the authors' response letter. Maybe some of the comments from here could be moved into the main text or the supplement."

and

"The authors have addressed my concerns, although the flow and narrative of the manuscript can be improved. For example, the authors addressed my comments regarding the lack of a table describing the parameters by including an external link in the manuscript to their code repository. This forces the reader to decipher and navigate the code repository website (github), thus creating an accessibility issue "

Once these minor changes are implemented, I will recommend the acceptance of the paper. I apologize for the longer than usual process, but now I think this can be done very quickly.

Best regards,

Gergely Röst

We look forward to receiving your revised manuscript.

Kind regards,

Gergely Röst

Academic Editor

PLOS ONE

Reviewers' comments:

Reviewer's Responses to Questions

**Comments to the Author**

1. If the authors have adequately addressed your comments raised in a previous round of review and you feel that this manuscript is now acceptable for publication, you may indicate that here to bypass the “Comments to the Author” section, enter your conflict of interest statement in the “Confidential to Editor” section, and submit your "Accept" recommendation.

Reviewer #1: (No Response)

Reviewer #2: All comments have been addressed

2. Is the manuscript technically sound, and do the data support the conclusions?

Reviewer #1: Yes

Reviewer #2: Yes

3. Has the statistical analysis been performed appropriately and rigorously? 

Reviewer #1: Yes

Reviewer #2: Yes

4. Have the authors made all data underlying the findings in their manuscript fully available?

Reviewer #1: Yes

Reviewer #2: Yes

5. Is the manuscript presented in an intelligible fashion and written in standard English?

Reviewer #1: Yes

Reviewer #2: Yes

6. Review Comments to the Author

Reviewer #1: The authors properly answered my questions in their response letter. My major concern about asymptomatic patients is now mentioned in the main text, with a reference to more details in the supplement, but the supplement was only minimally changed and the word asymptomatic does not even appear in the supplement. Thus some further update on the supplement and maybe on the main text could help to make this complete.

The results and the discussion sections are relatively short. It might have been useful to add a section on the limitations of the model as it stands now.

The above points are quite properly explained now in the authors' response letter. Maybe some of the comments from here could be moved into the main text or the supplement.

Reviewer #2: Dear Authors,

I would like to apologize for the delay in my review. I have found that all my comments have been addressed fully. Thank you.

7. PLOS authors have the option to publish the peer review history of their article (what does this mean?). If published, this will include your full peer review and any attached files.

Reviewer #1: No

Reviewer #2: No

---

## [Author Response · Author response to Decision Letter 1]

27 Jan 2021

Comment:

"The authors properly answered my questions in their response letter. My major concern about asymptomatic patients is now mentioned in the main text, with a reference to more details in the supplement, but the supplement was only minimally changed and the word asymptomatic does not even appear in the supplement. Thus some further update on the supplement and maybe on the main text could help to make this complete. The results and the discussion sections are relatively short. It might have been useful to add a section on the limitations of the model as it stands now. The above points are quite properly explained now in the authors' response letter. Maybe some of the comments from here could be moved into the main text or the supplement."

Answer:

We have updated both the manuscript and the supplement text with the discussion about the asymptomatic cases. 

We have also updated the Discussion section of the manuscript where we discussed the model limitations outlined earlier. 

The changes above are also emphasized in the tracked versions of the manuscript and of the supplement.

and

Comment:

"The authors have addressed my concerns, although the flow and narrative of the manuscript can be improved. For example, the authors addressed my comments regarding the lack of a table describing the parameters by including an external link in the manuscript to their code repository. This forces the reader to decipher and navigate the code repository website (github), thus creating an accessibility issue "

Answer:

Upon request, we have included an example of the parameters table in the supplement and referred the reader to the online repository for more detailed documentation about the tool. 

The changes above are also emphasized in the tracked versions of the supplement.

Comment:

Answer:

No changes with the previous submission and re-submission.

Comment:

Reviewer #1: The authors properly answered my questions in their response letter. My major concern about asymptomatic patients is now mentioned in the main text, with a reference to more details in the supplement, but the supplement was only minimally changed and the word asymptomatic does not even appear in the supplement. Thus some further update on the supplement and maybe on the main text could help to make this complete.

The results and the discussion sections are relatively short. It might have been useful to add a section on the limitations of the model as it stands now.

The above points are quite properly explained now in the authors' response letter. Maybe some of the comments from here could be moved into the main text or the supplement.

Answer:

We have updated both the manuscript and the supplement text with the discussion about the asymptomatic cases. 

We have also updated the Discussion section of the manuscript where we discussed the model limitations outlined earlier. 

The changes above are also emphasized in the tracked versions of the manuscript and of the supplement.

---

## [Editor Report · Decision Letter 2]

3 Feb 2021

Development of an interactive, agent-based local stochastic model of COVID-19 transmission and evaluation of mitigation strategies illustrated for the state of Massachusetts, USA

PONE-D-20-24277R2

Dear Dr. Gankin,

We’re pleased to inform you that your manuscript has been judged scientifically suitable for publication and will be formally accepted for publication once it meets all outstanding technical requirements.

Kind regards,

Gergely Röst

Academic Editor

PLOS ONE
---

## [Editor Report · Acceptance letter]

5 Feb 2021

PONE-D-20-24277R2 

Development of an interactive, agent-based local stochastic model of COVID-19 transmission and evaluation of mitigation strategies illustrated for the state of Massachusetts, USA 

Dear Dr. Gankin:

I'm pleased to inform you that your manuscript has been deemed suitable for publication in PLOS ONE. Congratulations! Your manuscript is now with our production department. 

Kind regards, 

on behalf of

Dr. Gergely Röst 

Academic Editor

PLOS ONE